# A New Method to Assess Fine-Scale Outdoor Thermal Comfort for Urban Agglomerations

**Dirk Lauwaet [1],\*, Bino Maiheu [1] , Koen De Ridder [1], Wesley Boënne [1] , Hans Hooyberghs [1], Matthias Demuzere [2,3] and Marie-Leen Verdonck [4]**

[1]  Vlaamse Instelling voor Technologisch Onderzoek (VITO), 2400 Mol, Belgium; bino.maiheu@vito.be (B.M.); koen.deridder@vito.be (K.D.R.); wesley.boenne@vito.be (W.B.); hans.hooyberghs@vito.be (H.H.)

[2]  Department of Geography, Ruhr-University Bochum, 44801 Bochum, Germany; matthias.demuzere@rub.de

[3]  Department of Environment, Laboratory of Hydrology and Water Management, Ghent University, 9000 Ghent, Belgium

[4]  Antea Group, 2000 Antwerpen, Belgium; marie-leen.verdonck@anteagroup.com

\*   Correspondence: dirk.lauwaet@vito.be

**Abstract:** In urban areas, high air temperatures and heat stress levels greatly affect human thermal comfort and public health, with climate change further increasing the mortality risks. This study presents a high resolution (100 m) modelling method, including detailed offline radiation calculations, that is able to efficiently calculate outdoor heat stress for entire urban agglomerations for a time period spanning several months. A dedicated measurement campaign was set up to evaluate model performance, yielding satisfactory results. As an example, the modelling tool was used to assess the effectiveness of green areas and water surfaces to cool air temperatures and wet bulb globe temperatures during a typical hot day in the city of Ghent (Belgium), since the use of vegetation and water bodies are shown to be promising in mitigating the adverse effects of urban heat islands and improving thermal comfort. The results show that air temperature reduction is most profound over water surfaces during the afternoon, while open rural areas are coolest during the night. Radiation shading from trees, and to a lesser extent, from buildings, is found to be most effective in reducing wet bulb globe temperatures and improving thermal comfort during the warmest moments of the day.

**Keywords:** thermal comfort; urban greening; urban heat island; UrbClim model; water bodies

## 1. Introduction

Rapid urban growth coupled with high population density increases the vulnerability of cities to extreme weather [1]. In cities, heat extremes are among the most important weather-related health hazards. Moreover, the effects of extreme heat are exacerbated by the presence of the urban heat island (UHI) [2]. This UHI is caused by a combination of the increased heat capacity of cities, anthropogenic heat sources, and the imperviousness of urban surfaces, which inhibit evaporative cooling [3–5]. Due to the UHI increment, cities are particularly vulnerable to heat waves, causing higher heat-related excess mortalities [6–8]. The risks of morbidity and mortality in urban areas are further increased by climate change, due to the increasing frequency of weather extremes [9,10].

In this context, integrating mitigation and adaptation measures can help avoid locking a city into counterproductive infrastructure and policies. Urban greenery has been proposed as an effective measure to mitigate the UHI and improve the urban microclimate [11–13]. Green areas are generally cooler than their surrounding built up areas, which is demonstrated by observational studies reporting instantaneous air temperature differences of 1 °C up to 7 °C [14,15]. Vegetation cools cities via shading, evapotranspiration, and alteration of the wind pattern [16]. The cooling intensity of, for example,

city parks, is often largest in the evenings and during the night (like the UHI) and tends to increase with park size [17]. During the afternoon, parks with extensive tree coverage tend to be cooler due to shading effects, while at night more open parks are cooler due to greater long-wave radiative cooling [18].

Similar to parks, water features have the potential to alleviate high urban temperatures through enhanced evaporation and reduced sensible heat fluxes [19]. A number of observational studies have shown that temperatures adjacent or downwind of water bodies are reduced by around 1–2 °C compared to surrounding areas [20,21]. On specific days during the afternoon, Murakawa et al. [22] observed 3–5 °C cooler temperatures near a wide river in Hiroshima, Japan. In contrast to urban parks where the cooling effect is most pronounced during the evening and the night, these studies suggest that open water effects are most pronounced during the day, because water bodies can maintain warmer temperatures at night due to the high heat capacity and thermal inertia of water [23].

However, air temperatures are only part of the story when assessing outdoor human comfort, as humidity, wind, and radiation also play a role [24,25]. During the day in summertime, radiation is even the single most important meteorological factor influencing the human energy balance [26]. Hence, by blocking solar radiation, trees can substantially improve thermal comfort, even if air temperature reductions are small. The relationship between urban greening and outdoor thermal comfort has been the subject of many modelling studies (e.g., [12,27,28]). At the same time, new urban modelling tools have been developed that focus on the introduction of green in urban design [29,30]. Most of these studies have been performed with high resolution models (e.g., ENVI-met) that can only perform simulations for limited areas in a city and for limited time periods.

In this study, we present an offline GIS-based post processing method, coupled to the urban boundary layer climate model UrbClim [31], that is able to calculate outdoor thermal comfort for an entire urban agglomeration and for time periods of several months to years. As a heat stress indicator, we applied the wet bulb globe temperature (WBGT), the ISO standard to quantify human thermal comfort [32]. The modelling method can easily be validated, which is done by performing an extensive measurement campaign in the city of Ghent, Belgium. As an example, the model is used to assess the effectiveness of urban vegetation and open water areas in reducing air temperatures and heat stress for the city of Ghent during a particularly warm summer day.

## 2. Materials and Methods

### 2.1. The UrbClim Model

The urban boundary layer climate model UrbClim [31] is designed to cover agglomeration-scale domains at a high spatial resolution, taken as 100 m for this study. The model consists of a land surface scheme, including simplified urban physics, which is coupled to a 3-D atmospheric boundary layer model. To ensure that the synoptic forcing is properly taken into account, the boundary layer model is tied to synoptic-scale meteorological fields through the lateral and top boundary conditions. The land surface scheme used in UrbClim is based on the soil–vegetation–atmosphere transfer scheme of De Ridder and Schayes [33], extended to account for urban surface physics. This urbanization is implemented by representing the urban surface as a rough impermeable slab, with appropriate values for the albedo, emissivity, thermal conductivity, and volumetric heat capacity. The main feature of the urbanization scheme is the inclusion of a parameterization of the inverse Stanton number, which is known to be much higher in urban areas [34,35]. A full description of the UrbClim model can be found in De Ridder et al. [31].

The spatial distribution of land cover types, needed for the specification of required land surface parameters, is taken from the reference land use map for Flanders, described in White et al. [36]. The percentage of urban land cover is attributed by applying the urban soil sealing raster data files that are distributed by the European environment agency. From the normalized difference vegetation index (NDVI) acquired by the MODIS instrument on-board the TERRA satellite platform, maps of

vegetation cover fraction were obtained. This fraction is specified as a function of the NDVI using a linear relationship proposed by Gutman and Ignatov [37], and then interpolated to the model grid. Model grid cells are divided into vegetation and bare soil (the complementary fraction) if they do not feature urban land use types. Where grid cells contain urban land use, the urban fraction, as derived from the soil sealing data, takes precedence over the fractional vegetation cover data, in case both sum to over 100%. In case they sum to less than that, the remaining fraction is assigned to bare soil. Terrain elevation data are taken from the GMTED2010 Dataset [38].

The UrbClim model has previously been validated regarding its energy fluxes, 2 m wind speeds, air temperatures, and urban–rural temperature differences for the cities of Antwerp, Brussels, and Ghent in Belgium, Toulouse in France, and Barcelona in Spain [31,39–41]. Also, the land surface temperatures in the UrbClim land surface scheme have already been validated with satellite data for the city of London [42]. In De Ridder [43], the urban parameterization was tested for the city of Paris, and the simulated land surface temperature compared favorably to observed values obtained from thermal infrared satellite imagery.

## 2.2. Outdoor WBGT Calculation

The UrbClim model has already been coupled offline to a building energy simulation model to calculate indoor heat stress, based on the WBGT [44]. Here, we present a method to calculate outdoor WBGT values from the UrbClim model results. Therefore, we follow the method of Liljegren et al. [45] to calculate the WBGT from standard meteorological variables, which is recommended for outdoor WBGT calculations in a review paper by Lemke and Kjellstrom [46]. The (outdoor) WBGT is the weighted sum of the natural wet bulb temperature $T_w$, the globe temperature $T_g$, and the dry bulb (ambient) temperature $T_a$:

$$WBGT = 0.7T_w + 0.2T_g + 0.1T_a \tag{1}$$

Separate calculation models for the natural wet bulb temperature and the globe temperature are used by Liljegren et al. [45], where all details on the calculations and required input parameters can be found.

In our modelling approach, hourly 2 m air temperatures, specific humidity, and wind speeds were taken from the UrbClim output data as input for the WBGT calculations. Downward solar radiation and surface pressure are also needed, and were taken from the ERA-interim re-analysis of the European centre for medium-range weather forecasting (ECMWF). Both these variables also serve as input data for the UrbClim model. To calculate the detailed amount of shade (either from buildings or trees) in a grid cell, shape files with building height and tree height were obtained from the city administration of Ghent. These files were converted to 1 m resolution raster files, and subsequently, for every hour of the period under study (see Section 2.3), the incoming solar radiation for the different solar zenith angles was calculated with the potential incoming solar radiation module of the system for automated geoscientific analyses (SAGA), an open-source geographic information system [47]. The results of these calculations were resampled to the UrbClim model grid, yielding fractional coverages of building shadow and tree shadow in every grid cell for every hour of the day. Finally, the fraction of the grid cells occupied by buildings was defined by resampling the building footprint raster to the model grid. We excluded this fraction from the calculations since we assume people do not stand regularly on their roof.

In the end, there were three types of locations in every grid cell: open, in the shadow of buildings, or in the shadow of trees. The open locations received the full amount of incoming solar radiation (both direct and diffuse, which are separately needed for the WBGT calculation), while for locations in the shadow of buildings, the direct fraction is set to 0 and only the diffuse fraction remains. In the shade of trees, the amount of direct and diffuse solar radiation was calculated according to the radiation transfer through the tree canopy scheme of De Ridder [48]. The overall WBGT value in a grid cell was taken as the weighted average of the respective WBGT values from these fractions.

Based on the approach outlined above, it is possible to obtain, in a fairly simple manner, outdoor heat stress values from general climate models, taking into account the detailed radiative shading effects of buildings and trees.

### 2.3. Experiment Setup

In order to evaluate the performance of the UrbClim model and the WBGT calculation, while also assessing the effectiveness of green and blue infrastructure in reducing outdoor heat stress, a dedicated measurement campaign was set up during the summer of 2015 in the city of Ghent. A measurement station was installed in the middle of a square of a public nature museum (Wereld van Kina, http://www.dewereldvankina.be/) in the city center during the months of July and August 2015. The station featured a WBGT sensor measuring 2 m (dry bulb) air temperatures, wet bulb and black globe temperatures, and relative humidity. The dry bulb temperature was measured using a highly accurate 43347 RTD temperature probe housed inside a fan-aspirated radiation shield, yielding a reported measurement uncertainty of only 0.1 °K. The relative humidity sensor has a reported measurement uncertainty of up to 5%.

During a particularly warm and sunny day (the 3rd of July 2015), two bikes were equipped with autonomous Onset HOBO U23-002 (http://www.onsetcomp.com/products/data-loggers/u23-002) temperature/humidity loggers at 2 m height and driven around the city center, through urban parks and to nearby rural areas, during the afternoon (12–16 h Local Time). The sensor tips were mounted in the same fan-aspirated radiation shields as the reference measurement station, using battery packs to power the fans during the trips. These measurements have a temporal resolution of 1 s. Furthermore, a floating fixed platform [49] was equipped with an automatic ventilated air temperature sensor (HOBO U23-002) at 2 m height to measure the cooling effect of a small lake near the city center, during the same time period as the bike measurements. Figure 1 shows the location and trajectories of all measurement devices.

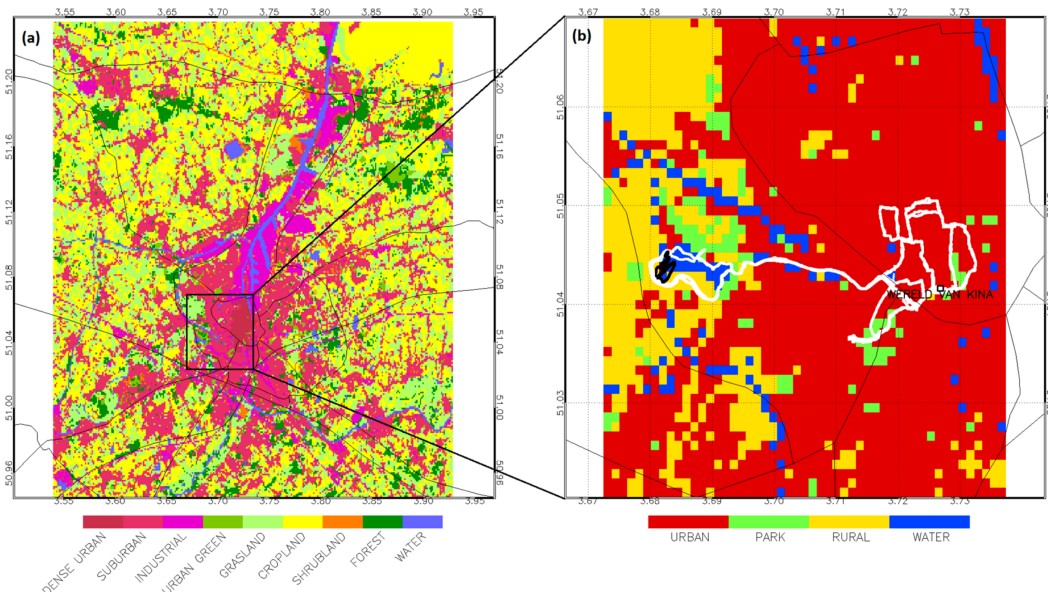

**Figure 1.** Extent and land use classification of the UrbClim model domain (**a**) and the focus area of this study around the measurement locations (**b**). The location of the observational station and the trajectories of the cycling (white) and boating (black) measurements are shown in the panel on the right.

Afterwards, the UrbClim model and the offline WBGT calculation were used to simulate the summer of 2015 for the wider agglomeration of Ghent, directly driven with meteorological data from the ERA-Interim re-analysis of the ECMWF, as was the setup in previous validation experiments [31,39–41]. The model domain is configured with 301 × 301 grid cells in the horizontal direction, using a spatial

resolution of 100 m. Figure 1 shows the extent of the UrbClim model domain and the land use in the domain. In the vertical direction, 20 levels are specified, with the first level 10 m above the displacement height, the resolution smoothly decreasing upward to 250 m at the model top located at 3 km height. This vertical discretization closely matches that of the ECMWF host model. The simulation is initialized on 1 May at 0000 LT, resulting in a two-month spin-up before the start of the analysis on 1 July, in order to ensure model equilibrium between external forcing and internal dynamics, especially in terms of soil variables. Initial soil temperature and soil moisture data are taken from the ERA-Interim re-analysis.

## 3. Results

### 3.1. Model Evaluation

Figure 2 shows the time series and error statistics of the measured and modelled 2 m air temperatures, dew point temperatures, and wet bulb globe temperatures at the observational station for the months of July and August 2015. There is a good correspondence between the measured and simulated air temperatures, with almost no bias, root mean square errors well below 2 °C, and a correlation coefficient over 0.9. These statistics are in line with previous validation results of the UrbClim model [31,39,41]. There is more variability in the comparison of the dew point temperatures, which is to some extend due to the larger uncertainty in the measurements. The measured dew point temperature is estimated from the air temperature and the relative humidity, of which the combined measurement errors can lead to an uncertainty well of over 0.5 °C. With this in mind, the error statistics are certainly reasonable. For the wet bulb globe temperatures, there is again a very good correspondence between measured and simulated temperatures, with a small positive bias, a root mean square error just above 1 °C, and a correlation coefficient of 0.95. There are a few days (e.g., 2 July) where the simulations deviate significantly from the observations, due to some observed cloudiness that is not picked up by the ERA-Interim re-analysis.

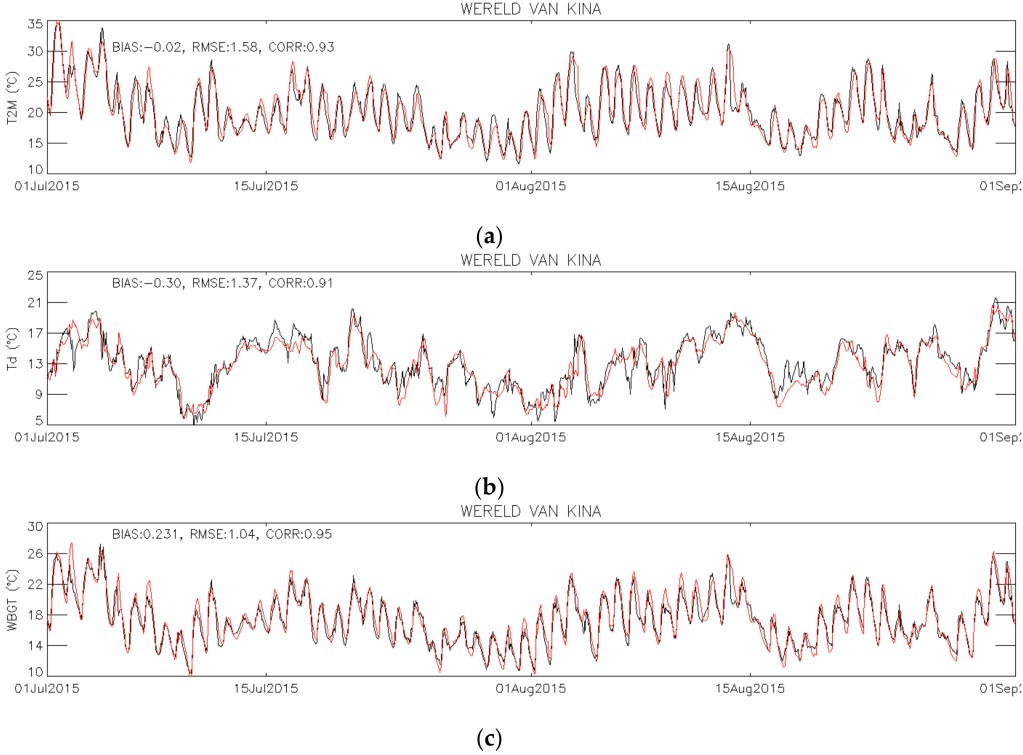

**Figure 2.** Time series of 2 m air temperatures (**a**), 2 m dew point temperatures (**b**), and wet bulb globe temperatures (**c**) for July and August 2015. Observations are in black, model results in red. The quantities given are the bias, root mean square error (RMSE), and correlation coefficient (CORR).

It is not straightforward to compare the simulated air temperatures to the mobile measurements. First of all, there is some local variability in land use that cannot be captured by the 100 m model resolution. Moreover, the model results have an hourly temporal resolution, whereas the observations have a temporal resolution of 1 s. To overcome these issues, we have grouped the measurements in four relevant land use classes (urban, park, rural, and water) and calculated the average value and standard deviation during a specific two hour period (14–16 h LT) of the afternoon of 3 July 2015. These values are compared to the modelled air temperatures for the same hours, taken from all grid cells of the focus area (Figure 1) for the respective land use classes. Table 1 shows the results of this comparison.

**Table 1.** Overview per land use class of the mean modelled and measured 2 m air temperatures, and their standard deviations (SD), for the afternoon (14–16 h Local Time) of 3 July 2015.

| Land Use | Model | | Measurements | |
|---|---|---|---|---|
| | Mean [°C] | SD [°C] | Mean [°C] | SD [°C] |
| Urban | 30.2 | 0.26 | 30.0 | 0.45 |
| Park | 30.0 | 0.28 | 29.9 | 0.85 |
| Rural | 30.1 | 0.29 | 29.0 | 0.36 |
| Water | 28.4 | 0.43 | 28.4 | 0.32 |

The average model results are very close to the observed values for all land use classes, except for the rural class. The simulated rural air temperatures in the afternoon are almost as high as the urban ones, but the measured rural air temperatures are 1 °C lower than the urban air temperatures, so the model seems to underestimate, to some extent, the cooling effect of the rural locations during the afternoon. The differences between the other land use classes are well captured by the model, with the air temperatures over water being almost 2 °C lower than the urban and park air temperatures. The standard deviations of the measurements are significantly larger than these of the model results, demonstrating the large local variability within each land use class, which is difficult to capture for a 100 m resolution model.

Overall, the model performs satisfactorily compared to the measurements.

## 3.2. Effect of Green and Blue Areas on Air Temperatures

On the 3rd of July 2015, the air temperatures in the city of Ghent rose quickly during the day due to sunny and calm conditions, reaching over 30 °C during the day and only cooling down slightly during the evening and the night. In Figure 3 the daily cycle of modelled 2 m air temperatures for the different land use classes is presented. The results show that during these types of days, air temperature is highest in the urban areas, while urban parks show limited or no cooling effect on air temperatures during midday hours. The measurements presented in Table 1 suggested that the rural areas just outside the city are around 1 °C cooler than the city center during these warm hours, while the model estimates this effect to be a lot smaller. Both the measurements and the model show that the lowest temperatures during the afternoon are clearly found over water surfaces, being around 2 °C cooler during the warmest moment of the day.

It is important to also investigate the differences in night time temperatures, since this is the moment that the UHI is the strongest and has the biggest potential impact on human health, because the warmer urban night-time temperatures limit the recuperation of city inhabitants from heat stress during daytime [50,51]. During the night, the urban air temperatures are warmest, with the air temperatures in parks and over water being around 0.5 °C cooler. The lowest air temperatures during the night are found in the rural areas, which are around 2 °C cooler due to the greater long-wave radiative cooling in the open fields and grasslands.

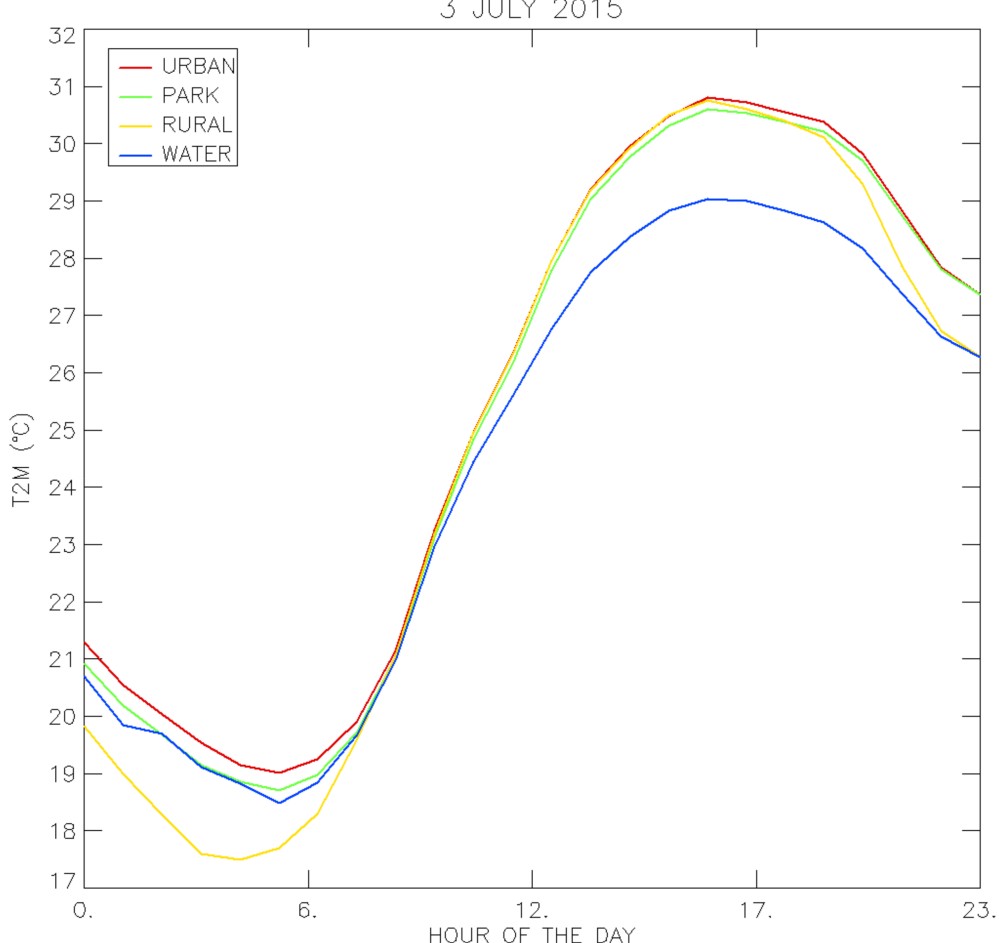

**Figure 3.** Daily cycle of modelled 2 m air temperatures with an hourly time resolution, averaged per land use class for the focus area.

### 3.3. Effect of Green and Blue Areas on Thermal Comfort

As explained in the introduction, it is important to also include humidity, wind speed, and radiation when assessing outdoor thermal comfort. Here, we apply the WBGT as a heat stress indicator, which is the ISO standard for human thermal comfort and takes these variables into account. In Figure 4, the daily cycle of the WBGT for the different land use classes is presented, differentiating between open and shaded locations.

Shading clearly plays a very important role regarding thermal comfort during the afternoon, when the differences in WBGT between all open locations are small (<1 °C), regardless of the land use. Even the water areas, where the air temperatures are several degrees cooler, have a WBGT value comparable to urban areas in the afternoon, due to the higher humidity, which offsets the cooler air temperatures. In urban areas, building shade provides a substantial cooling effect of 2 °C on the WBGT. The shade of trees in urban parks and rural areas has an even bigger cooling effect of around 3 °C. These are certainly the best places to avoid outdoor heat stress during a typical hot day in Ghent.

During the night, when there is no solar radiation, the WBGT values are in line with the air temperature results. Urban areas are warmest, with WBGT values in parks and over water being around 0.5 °C cooler. The coolest locations are the open rural areas, where minimal WBGT values are around 2 °C cooler than in urban areas due to the lower air temperatures there.

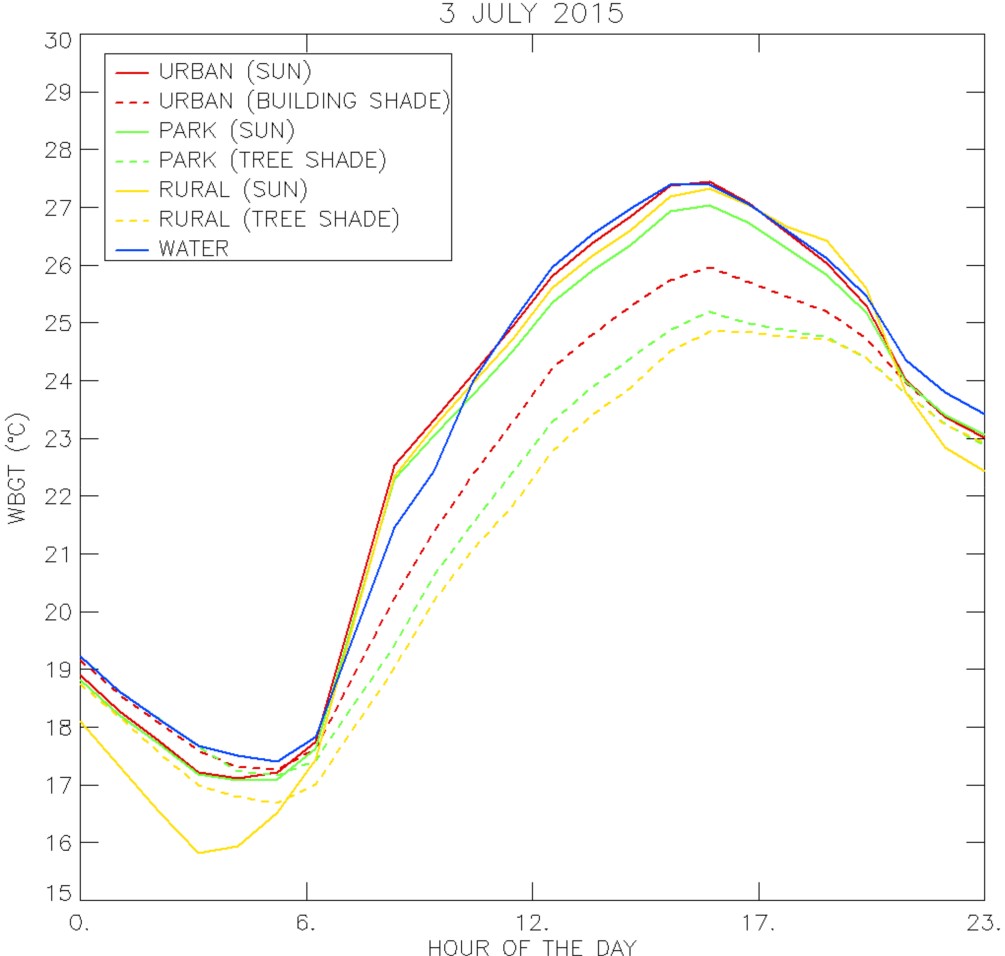

**Figure 4.** Daily cycle of modelled wet bulb globe temperatures with an hourly time resolution, averaged per land use class for the focus area. A distinction is made between unshaded (full lines) and fully shaded (dashed lines) areas.

## 4. Discussion and Conclusions

In this paper, we have presented an offline GIS-based post processing method, coupled to the urban climate model UrbClim, that can be used to calculate outdoor thermal comfort. The main advantage of our methodology is that we are able to calculate outdoor thermal comfort with a relatively high horizontal resolution for entire urban agglomerations and for long time periods (months to years), which is not possible with state of the art microclimate models. Moreover, the model can be easily validated with measurements, as we have demonstrated for the city of Ghent in Belgium, yielding satisfactorily results. As an example of the potential use of the model, the effectiveness of urban vegetation and open water areas in reducing air temperatures and heat stress for the city of Ghent during a particular warm summer day was assessed.

The modelling results regarding the effect of vegetation and water surfaces on air temperatures are in line with previous observational and modelling studies [20,21], with cooling effects of water surfaces up to 2 °C during daytime, and much smaller effects for night time temperatures when the water bodies maintain high water temperatures due to their high heat capacity. The cooling effect of park areas that is found in this study is small in comparison to reported observed values [12,13], which is probably due to the very small size of the city parks in Ghent, which, on top of that, feature a considerable amount of sealed surfaces. Related research studies on the effect of trees and water on outdoor heat stress have found similar results to the impact numbers reported here [27,28], with trees providing a substantial improvement to human thermal comfort by blocking solar radiation, even if air

temperature reductions are negligible. These types of scenario studies for urban areas (e.g., [52]) are performed with limited-area microclimate models such as ENVI-met that can only model a small part of a city for a few selected days.

Our methodology allows us to map outdoor thermal comfort for an entire city. As an example, Figures 5 and 6 show the daily maximum and minimum WBGT values on the 3rd of July for the city of Ghent, respectively. The results are limited to the city administrative area, since we only have the detailed building and tree height data for this area. The figures provide further evidence for the results discussed above, as the highest maximum WBGT values are found over open water and open urban areas, whereas the lowest maximum WBGT values are found in forested areas. The minimum WBGT map reflects the UHI situation for the city of Ghent with the warmest locations found in the city center and the urbanized areas around the city, and the coolest locations being the open rural areas outside the city.

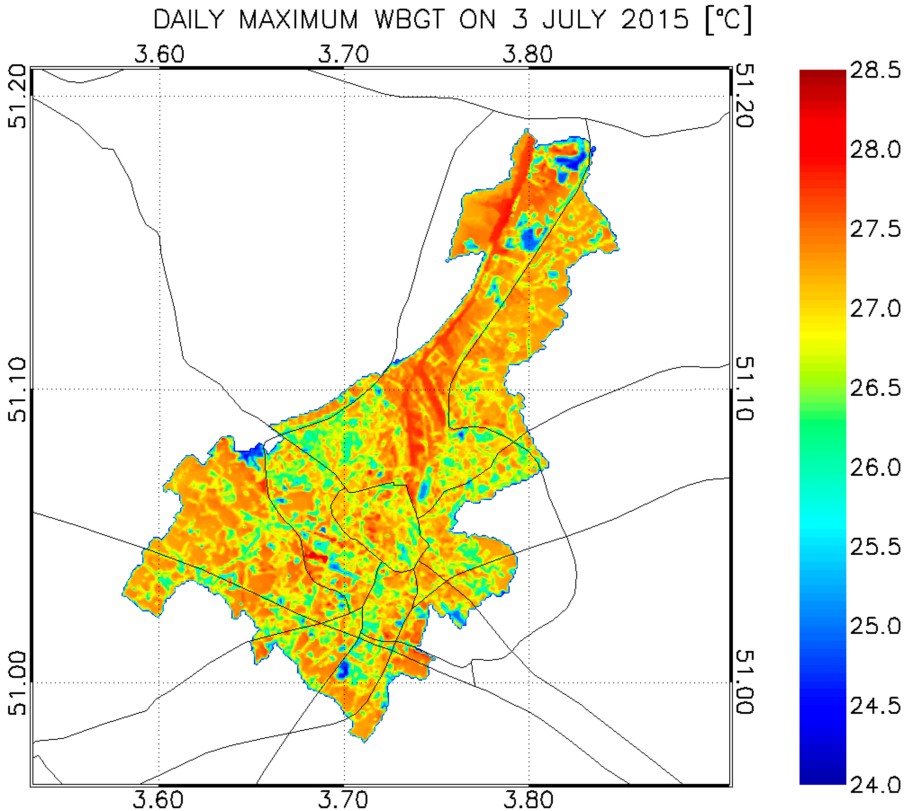

**Figure 5.** Map of the daily maximal wet bulb globe temperature for the city of Ghent.

Note that our analysis here focuses on a single hot day with calm and clear sky conditions. The reported cooling effects for the different land use types will differ and are lower when winds are stronger or more clouds are present. Also, in reality, the shading effect of trees will be dependent on the specific tree species and the health of the trees, which can be problematic in urban areas. Given these limitations and uncertainties, the results presented here are not suited for local risk assessment but should rather provide a coherent picture of potential outdoor heat stress benefits from water surfaces and green areas.

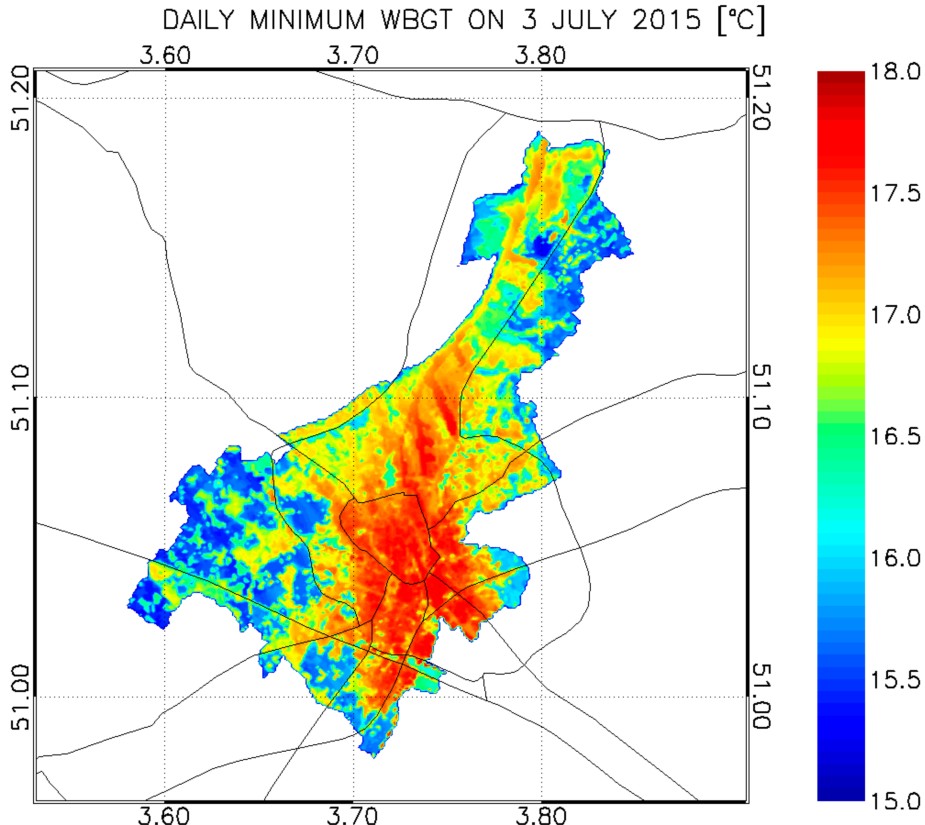

**Figure 6.** Map of the daily minimal wet bulb globe temperature for the city of Ghent.

**Author Contributions:** All authors contributed extensively to the work presented in this paper. K.D.R. supervised the project and is the creator of the UrbClim model. B.M. supervised the measurement campaign to which also K.D.R., W.B., M.D., and M.-L.V. contributed. H.H. was involved in the model creation and scripting. D.L. performed the model simulations, data analysis, wrote the manuscript, and drafted the figures. All authors discussed the results and implications and commented on the manuscript at all stages. All authors have read and agreed to the published version of the manuscript.

**Funding:** This research has been performed in the framework of the CORDEX.be project, funded by the Belgian Science Policy Office (BELSPO) under contract number BR/143/A2. Furthermore, the work described in this paper has received funding from the European Community's 7th Framework Programme under Grant Agreements Nos. 308497 (RAMSES) and 308299 (NACLIM) and from the European Union's H2020 Research and Innovation Programme under Grant Agreement No. 73004 (PUCS/Climate-fit.city). Marie-Leen Verdonck was funded by the Belgian Federal Science Policy Office, as part of the UrbanEARS project. Matthias Demuzere was funded by the FWO (Fund for Scientific Research) of the Flemish regional government.

**Conflicts of Interest:** The authors declare no conflict of interest.

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
