# Peer review of "A New Method to Assess Fine-Scale Outdoor Thermal Comfort for Urban Agglomerations"

_climate, doi:10.3390/cli8010006_

Round 1
Reviewer 1 Report
The work presents an interesting discussion, for a real case, of the advantages in terms of well-being offered by the application of blue-green solutions.In fact, the results clearly indicate the effects of the presence of areas with free water and those due to vegetable coverings also evaluating the period of the day in which these effects occur.
This is a fundamental indication for the planners who can thus have an expert guide in the formulation of the architectural-urban layout.
The study also offers an interesting starting point for discussion relating to continuous monitoring of well-being to protect urban populations.
The study is of interest for a wide audience, well prepared and discussed.
I suggest it for publication as it is.
Author Response
The authors would like to thank the reviewer for his/her positive feedback.
Reviewer 2 Report
Attached to the file.

Author Response
The authors would like to thank the reviewer for his/her constructive comments. Please see the attachment to find our responses to the detailed comments.

Reviewer 3 Report
The authors have presented an interesting study where they have used an urban climate model to obtain data for an large area and then computed the thermal comfort for the urban space. They validated their model with measurement during the summer time. They finally showed that they were able to produce a map of the comfort for an urban area for specific days. The authors should make some changes before the article if acceptable for publication.
They need to include in their introduction why they believe their work is novel. Previous models have already included some of the aspect they are mentioning. In Figure 2, they need to explain or give more details as to why the dew point temperature is the worst performing (although within reasonable values). Can the authors please modify Figure 3? It is very difficult to make sense of the curves in particular in the crowded part of the figure where several lines and the error bars overlap.
Other minor comments / suggestions can be found attached.

Author Response
Reviewer 3
The authors have presented an interesting study where they have used an urban climate model to obtain data for an large area and then computed the thermal comfort for the urban space. They validated their model with measurement during the summer time. They finally showed that they were able to produce a map of the comfort for an urban area for specific days.
The authors would like to thank the reviewer for his/her positive feedback.
They need to include in their introduction why they believe their work is novel. Previous models have already included some of the aspect they are mentioning.
The authors agree with the reviewer that this point could be strengthened. Therefore, the last part of the introduction has been rewritten, as are the conclusions. Compared to related models, the advantage of our methodology is that we are able to map outdoor thermal comfort for entire urban agglomerations, and for long time periods (months to years). This potentially allows for climate change simulations. Furthermore, the indicators that are modelled (UHI and WBGT) can be easily validated, as demonstrated in the paper.
In Figure 2, they need to explain or give more details as to why the dew point temperature is the worst performing (although within reasonable values).
More explanation is given in the revised paper, where we acknowledge that there is more variability in the comparison of the dew point temperatures. This is to some extend due to the larger uncertainty in the measurements, as the measured dew point temperature is estimated from the air temperature and the relative humidity, of which the combined measurement errors can lead to an uncertainty well over 0.5°C. With this in mind, the error statistics are certainly reasonable.
Can the authors please modify Figure 3? It is very difficult to make sense of the curves in particular in the crowded part of the figure where several lines and the error bars overlap.
The authors agree with the reviewer that Figure 3 was not very readable. Therefore, we have moved the comparison with the observations to a separate table (Table 1).
Other minor comments / suggestions can be found attached.
We have added the requested references and concluding paragraph.